# Phytochemical Profile of *Foeniculum vulgare* Subsp. *piperitum* Essential Oils and Evaluation of Acaricidal Efficacy against *Varroa destructor* in *Apis mellifera* by In Vitro and Semi-Field Fumigation Tests

**DOI:** 10.3390/vetsci9120684

**Published:** 2022-12-09

**Authors:** Roberto Bava, Fabio Castagna, Ernesto Palma, Vincenzo Musolino, Cristina Carresi, Antonio Cardamone, Carmine Lupia, Mariangela Marrelli, Filomena Conforti, Paola Roncada, Vincenzo Musella, Domenico Britti

**Affiliations:** 1Department of Health Sciences, University of Catanzaro Magna Græcia, 88100 Catanzaro, Italy; 2Interdepartmental Center Veterinary Service for Human and Animal Health, University of Catanzaro Magna Græcia, CISVetSUA, 88100 Catanzaro, Italy; 3Department of Health Sciences, Institute of Research for Food Safety & Health (IRC-FISH), University of Catanzaro Magna Græcia, 88100 Catanzaro, Italy; 4Nutramed S.c.a.r.l., Complesso Ninì Barbieri, Roccelletta di Borgia, 88021 Catanzaro, Italy; 5Pharmaceutical Biology Laboratory, Department of Health Sciences, Institute of Research for Food Safety & Health (IRC-FISH), University of Catanzaro Magna Græcia, 88100 Catanzaro, Italy; 6Mediterranean Ethnobotanical Conservatory, Sersale (CZ), 88054 Catanzaro, Italy; 7National Ethnobotanical Conservatory, Castelluccio Superiore, 85040 Potenza, Italy; 8Department of Pharmacy, Health and Nutritional Sciences, University of Calabria, 87036 Rende, Italy

**Keywords:** *Varroa destructor*, *Apis mellifera*, IPM (integrated pest management), *Foeniculum vulgare* subsp. *piperitum* essential oils, phytochemical profile, fumigation toxicity, in vitro and semi-field tests

## Abstract

**Simple Summary:**

Essential oils (EOs) are promising tools for controlling *Varroa destructor* mites. The extreme volatility that characterizes these compounds makes it possible to achieve acaricide action without the parasites coming into contact with the drug. Despite this particularity, relatively few studies have analyzed the effectiveness of fumigation as opposed to contact efficacy. In this study, the efficacy of the EO of *Foeniculum vulgare* sbps. *piperitum* (epigeal part) and its fractions (leaves, achenes and flowers) was evaluated through in vitro and semi-field experiments. Fennel EO and its fractions showed moderate efficacy using techniques that only allowed evaporation. In particular, in in vitro tests, at the highest concentration and exposure time, the whole plant and the leaf, achene and flower fractions returned acaricidal efficacies of 56% (flowers), 52% (achenes), 64% (leaves) and 68% (whole plant). In the semi-field trials, the EO of the whole plant at the highest concentration caused the death of 53% of the mites exposed to the vapours.

**Abstract:**

Varroatosis is an important parasitic disease of *Apis mellifera* caused by the mite *Varroa destructor (V. destructor)*. The parasite is able to transmit numerous pathogens to honeybees which can lead to colony collapse. In recent years, the effectiveness of authorized drug products has decreased due to increasing resistance phenomena. Therefore, the search for alternatives to commercially available drugs is mandatory. In this context, essential oils (EOs) prove to be a promising choice to be studied for their known acaricide properties. In this research work, the acaricide activity of EO vapours isolated from the epigeal part (whole plant) of fennel (*Foeniculum vulgare* sbps. *piperitum)* and its three fractions (leaves, achenes and flowers) against *V. destructor* was evaluated. The effectiveness of fumigation was studied using two methods. The first involved prolonged exposure of mites to oil vapour for variable times. After exposure, the five mites in each replicate were placed in a Petri dish with an *Apis mellifera* larva. Mortality, due to chronic toxicity phenomena, was assessed after 48 h. The second method aimed to translate the results obtained from the in vitro test into a semi-field experiment. Therefore, two-level cages were set up. In the lower compartment of the cage, a material releasing oil vapours was placed; in the upper compartment, Varroa-infested honeybees were set. The results of the first method showed that the increase in mortality was directly proportional to exposure time and concentration. The whole plant returned 68% mortality at the highest concentration (2 mg/mL) and highest exposure time (48 h control), while the leaves, achenes and flowers returned 64%, 52% and 56% mortality, respectively. In the semi-field experiment, a concentration up to 20 times higher than the one used in the in vitro study was required for the whole plant to achieve a similar mite drop of >50%. The results of the study show that in vitro tests should only be used for preliminary screening of EO activity. In vitro tests should be followed by semi-field tests, which are essential to identify the threshold of toxicity to bees and the effective dose to be used in field studies.

## 1. Introduction

Parasitosis caused by the *Varroa destructor* (*V. destructor*) mite is the main threat to the survival of honeybees (*Apis mellifera)* [1]. The parasite exerts a resource-depleting action on adult bees and larval forms [2]. This deprivation is associated with metabolic changes and immune stress due to the inoculation of numerous viruses and other pathogens that *V. destructor* transfers during its meal. The life cycle of the parasite comprises two phases: a dispersal phase and a reproductive phase. The first phase allows the mite to spread to other colonies and thus propagate the infestation [3]. Although useful for spreading the pest, this brief external exposure makes the mite susceptible to pharmacological treatments used for control purposes. There are several pharmaceutical preparations authorized for *Varroa* control. The products most commonly used to treat infected colonies are those based on organic acids (oxalic acid and formic acid), essential oils (e.g., thymol), organophosphates (e.g., coumaphos), pyrethroids (e.g., flumethrin and fluvalinate) and formamidine (Amitraz) [4]. Despite this armamentarium, misuse attributable to the administration of incorrect doses and/or the frequent use of the same active ingredient has favoured the development of mites resistant to the common acaricides [5,6,7,8,9]. The effectiveness of many molecules has therefore decreased. In this scenario, the scientific community is moving to embrace the approach that goes by the name of integrated pest management (IPM). This approach combines strategies such as the use of low-impact pharmaceutical preparations, the prudent use of drugs and staging of the degree of infestation. These strategies must be combined with preventive measures and biomechanical control techniques [4,10].

Given their characteristics, essential oils (EOs) can fit firmly into this context. Scientific research on pharmacological treatments based on EOs and plant mixtures in various animal species has seen a particular ferment [5,11,12,13,14,15]. EOs are complex mixtures of volatile secondary metabolites extracted and isolated from aromatic plants [16]. These metabolites are present in various plant parts (e.g., stems, buds, flowers, leaves, seeds, fruit, roots and bark), within specialized cells or glands [17,18]. In addition to attracting pollinators, plants employ these metabolites to protect themselves from predators and parasites [19]. Since ancient times, people have recognized that EOs possess antiparasitic, antiviral, antibacterial and antifungal activities. For example, recent studies have shown the therapeutic efficacy of EOs not only against *V. destructor* but also against the causative agent of chalkbrood disease, *Ascosphaera apis* [20]. The pharmaceutical activity of EOs is due to the active ingredients they contain. About 7000 different constituents have been identified, the most abundant of which are monoterpenes and sesquiterpenes. The variegated molecular composition makes the development of drug resistance phenomena complicated [21]. For this reason, if included in a therapeutic program that includes drug rotation, these compounds should be useful in limiting resistance. Furthermore, their natural origin may contribute to lower toxicity towards non-target organisms and the environment [22]. Although there have been several studies that have attested to the acaricidal efficacy of EOs by contact, few have investigated fumigant action [23]. EOs are volatile compounds and exert their main actions by evaporation. To be effective, a significant amount must evaporate until saturation of the air in the hive occurs and the required drug dose is reached.

The aim of this study was to evaluate the acaricidal activity by fumigation of the EO of the epigeal part (whole plant) of fennel (*Foeniculum vulgare* sbps. *piperitum)* and its fractions (leaves, achenes and flowers). The study offered the opportunity to investigate the mechanism of action of the phytocomplex in order to better define the synergy between the groups of molecules. Another important point that this research work intended to clarify was the best method for translating laboratory results into the field. There are no common methods for extrapolating laboratory achievements in the field. Therefore, we proposed to reveal the pros and cons of the in vitro and semi-field tests. In this case, the EO of fennel was the means that allowed us to make our considerations.

Fennel is a Mediterranean herbaceous shrub that arises spontaneously, especially in Central and Southern Italy, in the so-called Mediterranean scrub [24]. *Foeniculum* spp. EO is known to contain compounds such as terpenes, coumarins, flavonoids and sterols, mainly from trans-anethole, fenchon, estragon and α-fellandrene [25]. These constituents have shown pharmacological activity against mites, diptera, fleas, lepidoptera and ticks [26,27,28]. The presence and concentration of EO constituents is variable and mainly depends on the phenological stage and the geographical area in which the plant or its parts are harvested, as well as the extraction method. Furthermore, these compounds are present in different concentrations in the various parts of the plant (stems, buds, flowers, leaves, seeds, fruit, etc.). For all these reasons, the choice fell on this botanical species, and the scientific investigation was initiated.

## 2. Materials and Methods

The acaricidal efficacy and toxicity tests were conducted at the Interdepartmental Center for Veterinary Services, for Human and Animal Health (CISVetSUA), University “Magna Graecia” of Catanzaro, in June 2022. Mites and honeybees were employed for the operations.

The mites and honeybees used for the toxicity tests came from three apiaries in Calabria Region (Southern Italy). The hives had not been subjected to acaricidal treatments in the six months prior to the study. For the experimental purposes, some frames where the male brood had been reared were transferred from the hives to the laboratory. The male brood can supply a greater quantity of parasites, being preferred by *Varroa*, which finds greater reproductive success with it.

In the laboratory, under controlled conditions of temperature (24–25 °C), each brood cell was deprived of its operculum and examined for mites. If detected, adult female mites were taken from larvae and non-pigmented pupae and transferred onto a Petri dish using a tiny paintbrush.

Previously, fifth-stage larvae and/or bee pupae had been placed in the Petri dishes. This measure was useful to avoid causing nutritional stress to the collected parasites during the time necessary for the recovery of an adequate number. Before each test, mites that seemed abnormal were not included and were discarded because they could have died more easily, falsifying our toxicity tests. Once the number of mites was adequate for the preparation of the experimental replicates, the trials began. The honeybees used for the tests came from a now nascent brood honeycomb that was transported to the lab, where it was placed in an incubator.

In the incubation conditions, the adults had the possibility to come out. Once the bees had emerged, they were previously inspected for any *Varroa* on their bodies. After verifying the absence of parasites, the bees were placed in the experimental cages in groups of 20.

### 2.1. Plant Material

The whole plants (epigeal parts) and relative fractions (leaves, achenes and flowers) were collected in Calabria (Southern Italy) from the natural environment at an average elevation of 400 m above sea level. The collection times were as follows: leaves were harvested in June, the whole plant in August, and flowers and achenes in September and October, respectively.

Dr. Vincenzo Musolino (Pharmaceutical Biology Laboratory, Department of Health Sciences, Institute of Research for Food Safety & Health, University of Catanzaro Magna Graecia) and Dr. Carmine Lupia (Mediterranean Ethnobotanical Conservatory, Sersale (CZ), Italy) verified the taxonomic identity. The researchers also indicated the most favourable period for harvesting the plant and the three fractions used in the study.

The voucher was deposited in the Mediterranean Ethnobotanical Conservatory, Sersale (CZ), 88054 Catanzaro, Italy, under the number 55, section Apiaceae family.

### 2.2. Extraction Techniques and Phytochemical Profile and Gas Chromatography–Mass Spectrometry (GC–MS) Analyses

EOs were obtained from fresh samples, which were washed and extracted with steam distillation for 2 h, using a Clevenger-type apparatus (Albrigi Luigi, Verona, Italy). The obtained essential oils were then dried over anhydrous sodium sulfate and stored at +4 °C.

The following amounts of plant material were extracted: 2100.00 g of the whole plant, 2209.90 g of leaves, 1157.82 g of achenes and 761.66 g of flowers. The obtained yields were equal to 0.54, 0.13, 0.73 and 0.68% *w*/*w*, respectively.

For the chemical characterization of the EOs, analyses were carried out using a Hewlett–Packard 6890 gas chromatograph equipped with a 100% dimethylpolysiloxane SE-30 capillary column (30 m × 0.25 mm, 0.25 μm film thickness). The instrument was coupled to a Hewlett–Packard 5973 mass spectrometer. Helium was used as a carrier gas (linear velocity = 0.00167 cm/s), and analyses were performed using a programmed temperature from 60 to 280 °C (rate = 16 °C/min). The column inlet was set at 250 °C. The operating parameters were as follows: ion source, 70 eV; ion source temperature, 230 °C; electron current, 34.6 μA; vacuum, 10–5 torr. For the chemical characterization of EOs, GC retention times were compared to those of available standards, and for the mass spectra a Hewlett–Packard 6890 gas chromatograph equipped with a 100% dimethylpolysiloxane SE-30 capillary column was used. The instrument was coupled to a Hewlett–Packard 5973 mass spectrometer. To identify the constituents of EOs, the spectra obtained were compared with those deposited in the Wiley Mass Spectral Database of the GC–MS system [29].

### 2.3. In Vitro Toxicity Test

In agreement with Castagna et al. (2022), a cotton wool ball was inserted into the inner part of a 2 mL Eppendorf tube cap [11]. With the help of a small brush, for each experimental replicate, five adult female *Varroa* mites were transferred to the conical bottom of the Eppendorf tube. To avoid direct contact between the mites and the cotton wool, a piece of tulle was inserted into the Eppendorf tube. Subsequently, the cotton ball was soaked with 40 µL of a solution of distilled water and EO and the tube was hermetically sealed. The concentrations used were 1 mg/mL and 2 mg/mL. The above operations were carried out in the laboratory at a room temperature of 24–25 °C.

The samples prepared in this way were inserted in an incubator, maintaining a constant temperature of 34 °C and a relative humidity of 65%. *V. destructor* mites were exposed to the EO vapours for 15, 30, 45 and 90 min.

For the negative control, replicates of mites exposed to a cotton ball soaked with the same amount of distilled water were set up. After each interval, the mites were observed under the stereomicroscope and mortality was verified. Mites that did not move when probed with a fine paintbrush were considered dead; mites that moved one or more legs were classified as inactive. Both dead and inactive mites were classified as neutralized. Live mites from this first step were moved onto a Petri dish containing one honeybee larva for every replicate.

Subsequently, the dishes were returned to the incubator for 48 h at the same humidity and temperature. This second step allowed us to verify any possible mortality phenomena referable to chronic toxicity. In total, five experimental replicates were set up for each concentration, time of exposure and negative control.

### 2.4. Semi-Field Toxicity Test

As in Zheguang Lin et al.’s study (2019) [23], to verify the efficacy of the EO vapours, two-level cylindrical cages with heights of 10 cm and diameters of 9 cm were prepared.

The two floors of the cage were distinct and physically separated from each other.

Twenty adult honeybees and ten mites were placed in the upper part of the cage. The honeybees were from nascent brood combs taken from healthy colonies and treated against *V. destructor,* which had been brought into the laboratory and placed in the incubator to allow the bees to emerge.

Once in the cage, the honeybees were fed ad libitum with 50% sugar syrup. Subsequently, a filter paper was inserted in the lower compartment of the cage and soaked with oil solution (EO diluted in distilled water).

The volume of the cylinder was saturated by soaking a 2 × 2 cm strip of Whatman No. 1 with the different EO concentrations. The filter paper was inserted into the lower compartment of the fumigation chamber. Four different dosages were used. Precisely, the Whatman filter paper (2 × 2 cm) was treated with 1 mL of aqueous solution of EOs of the leaves, flowers and whole fennel plant. The concentrations ranged from 0 to 5%.

Five replicates were established for each concentration. The detachment of the mite from the host was only considered valid if the adult honeybees were alive at the conclusion of the test.

The behaviour and mortality of bees and mites were monitored for 1 h and established 48 h after treatment.

### 2.5. Honeybee Workers: Toxicity Evaluation

As in the fumigation semi-field tests for mites, twenty adult honeybees obtained by the same method were exposed to the EO vapours. Two-tiered cylindrical cages (height = 10 cm, diameter = 9 cm) equipped with troughs, filled with a 50% solution of sucrose and water, were set up. The honeybees were placed in the upper chamber of the cage and an EO-soaked filter paper was inserted in the lower chamber [23].

Subsequently, all cages were transferred to the incubator at a constant temperature of 34 °C and 65% relative humidity. The bees, under these conditions, were monitored for up to 48 h and any abnormal findings (e.g., asthenia, tremors and nervous disorders) and mortality were noted. Concentrations of 5 to 9% were used for toxicity tests. Toxicity tests were performed in three replicates.

### 2.6. Statistical Analysis

Data analysis was conducted using GraphPad PRISM (version 9.3.1, GraphPad Software Inc., La Jolla, CA, USA). The results are expressed as means ± SEMs. The Shapiro–Wilk test was used to assess normality. The data from two distinct groups were compared using the Mann–Whitney test.

JASP was utilized to perform a repeated measures ANOVA to compare the treatments at various concentrations (version 0.16.3, JASP Team, University of Amsterdam, Amsterdam, Netherlands). Mauchly’s test was used to verify the sphericity assumption. For pairwise comparisons, Holm’s post hoc test was used. Statistics were judged significant for values with *p* = 0.05 or above.

## 3. Results

### 3.1. Phytochemical Profiles

The phytochemical profiles of the EOs of *F. vulgare* subsp. *piperitum* are shown in the table below (Table 1). Overall, the phenylpropanoids estragole and anethole and the oxygenated monoterpene fenchone were the most abundant components. Anethole was the main phytochemical compound identified in the essential oil from the whole plant (49.90%), and it was detected in the other three samples in percentages ranging from 24.16% (achene EO) to 29.18% (leaf EO). Both estragole and fenchone were more abundant in the achene EO, with percentages equal to 33.40% and 18.48%, respectively.

### 3.2. Fumigant In Vitro Toxicity

Fumigant toxicity was assessed to evaluate the acute and subacute toxicity of EOs. As shown in Figure 1 and Table 2, the acute toxicity was time- and dose-dependent. However, in vitro acute toxicity was not statistically significantly impacted, according to the repeated measures ANOVA data, for the concentrations of 1 mg/mL (F(6.262,33.398) = 0.787, *p* = 0.591) and 2 mg/mL (F(9,48) = 0.916, *p* = 0.520).

The highest mortality rates were reached after 90 min only with the EOs isolated from the whole plant and leaf extracts at 1 mg/mL (*p* < 0.05 and *p* < 0.01 respectively) and 2 mg/mL (*p* < 0.001) (Table 2).

The subacute toxicity of the EOs was assessed after 24 and 48 h following the incubation of the remaining mites and using a honeybee larva as food (Figure 2 and Table 3). Mortality associated with subacute toxicity resulted from a longer duration of exposure to the EO vapours, which allowed a better control of the infestation rate. Indeed, a check at 24 and 48 h allowed us to highlight a significant increase in the mortality percentages at the concentrations of 1 mg/mL and 2 mg/mL.

At the concentration of 1 mg/mL, the mortality significantly increased for the EOs isolated from the whole plant (Figure 2) and leaves (Figure 2) after a time of 24 h. Moreover, as shown in Figure 2, after a time of 48 h, the mortality significantly increased for the EOs isolated by whole plant (*p* < 0.05), flowers (*p* < 0.05), achenes (*p* < 0.05) and leaves (*p* < 0.01). These results obtained after initial exposures may be related to chronic toxicity phenomena.

Similarly, at the concentration of 2 mg/mL, the number of neutralized mites significantly increased for the EOs isolated from the whole plant, flowers, achenes and leaves after verifications at 24 h and 48 h (Figure 2).

### 3.3. Fumigant Semi-Field Toxicity

For the fumigant semi-field toxicity experiment, four gradually increasing concentrations (10, 20, 30 and 40 mg/mL) of the EOs were tested. The purpose of the tests was to assess whether mites were detached from the bees’ bodies in the set-up cages. At 10 mg/mL, no detachment of the parasites at 24 h and 48 h was observed. Starting from 20 mg/mL, the detachment of mites was observed.

Furthermore, repeated measures ANOVA showed that a statistically significant effect on the detachment of parasites was observed after 24 h (F(9,24) = 4.576, *p* < 0.01) and 48 h of the exposition (F(9,24) = 7.884, *p* < 0.001).

Specifically, after 24 h of exposition, the detachment of mites was dose-related, as shown in the Figure 3A. The EO obtained from the whole plant showed the highest effectiveness at a concentration of 40 mg/mL (53.33%; Table 4).

Moreover, comparisons at specific concentrations showed that at 30 mg/mL the EO obtained from the whole plant was more effective than the EO obtained from the achenes (*p* < 0.05). Finally, at 40 mg/mL, the EO obtained from the whole plant was more effective compared to those obtained from achenes (*p* < 0.001), flowers (*p* < 0.001) and leaves (*p* < 0.001) (Figure 3A, Table 4).

Similarly, the detachment of mites after 48 h of exposition was dose-related (Figure 3B), and the highest percentage of detachment was reached using the EOs obtained from the whole plant at concentrations of 30 mg/mL (40%) and 40 mg/mL (53.33%) and from the leaves at a concentration of 40 mg/mL (36.67%) (Table 4).

The effectiveness of the EO obtained from the whole plant was significantly higher than those obtained from achenes (*p* < 0.001), flowers (*p* < 0.001) and leaves (*p* < 0.01) at 30 mg/mL (Figure 3B, Table 4). At 40 mg/mL, the EO obtained from the whole plant was significantly more effective than those obtained from flowers (*p* < 0.001) and achenes (*p* < 0.001), and the EO obtained from leaves showed a significantly higher effectiveness compared to the EOs obtained from achenes (*p* < 0.001) and flowers (*p* < 0.05) (Table 4, Figure 3B).

### 3.4. Honeybee Toxicity Evaluation

From a concentration of 7% (70 mg/mL), the oil of the whole plant proved to be toxic to the honeybees. In the experimental replicates, an average mortality of 55% of the subjects was observed after exposure to vapours for 24 h, which increased to 80% after 48 h.

At 24 h, a concentration of 80 mg/mL was responsible for the death of 70% of the exposed subjects, increasing to 86% after 48 h.

At 9% (90 mg/mL), the mortality of the subjects was total after 24 h of exposure to vapours. For the fractions, at a concentration of 7%, mortalities of 75%, 71.6% and 68.3% of the subjects after 48 h of exposure were recorded for the leaves, flowers and achenes, respectively. In the same order, at a concentration of 8%, mortality rose to 86%, 83% and 78.3%.

Finally, after 48 h, 9% of all the caged honeybees were dead.

## 4. Discussion

Numerous scientific studies have confirmed the effectiveness of EOs against a variety of parasites and pests of food products. Particular efforts have been made against parasites of stored products [30,31,32]. However, there is a lack of studies on lice, ticks and mites [33,34,35]. The present study is part of this line of research. EOs are frequently characterized by the presence of two or three components at quite high concentrations (20–70%) that stand out from the others present in traces [36]. The main components belong to the groups of terpenes, terpenoids and sesquiterpenes. Those present at lower percentages are aromatic and aliphatic constituents of low molecular weight. In our study, monoterpene, monoterpenoid and sesquiterpene hydrocarbons were identified in the oils. Monoterpenes, in particular, make up the majority of the EOs’ volatile components. Depending on the method used to verify toxicity, the effects of the constituents may vary. For example, 1,8-cineole, a main component of eucalyptus, rosemary and tea tree oils, has shown little efficacy in biological contact tests against *Sitophilus oryzae* and *Tribolium castaneum* insects; otherwise, it has a good toxic action when used for fumigation [37]. However, geraniol exhibits the exact opposite tendency, having high contact toxicity but minimal fumigant activity [37]. Due to their effectiveness as fumigants, several monoterpenes may be crucial in the control of ectoparasites. One piece of evidence that emerged from our experimental tests is the strong fumigant efficacy of the oils extracted from the whole plant compared to other fractions. Therefore, all components work together to produce most biological effects, and the most abundant chemicals have the greatest acaricidal effect. Analysis of the phytochemical profiles, however, revealed more abundant chemicals to which the greatest acaricide effect can be ascribed. As evidenced by other research, the acaricidal effect of fennel oil can be related to its high anethole amount [27], which is present in much higher concentrations (percentages) in the EO of the whole plant (49.90) than in the EOs of the leaves (29.18), achenes (24.16) and flowers (27.40). Considering that only vapours were tested, it can be concluded that anethole showed good acaricide action. However, anethole is not free from toxicity towards non-targets.

Our results are in line with the study of Haddadi (2021) on the activity of anethole in relation to honeybees [38] and Sabahi et al.’s (2018) study on *V. destructor* and workers and larvae of *A. mellifera* [39]. According to the first study’s findings, anethole can cause honeybees to experience oxidative stress, which can have an impact on the insects’ ability to survive, while in the second study the toxicity to worker honeybees was consistently greater than that of Tagetes oil. At the same time, the toxicity for *V. destructor* was dose-dependent and higher than that of Tagetes oil. In the study by Hybl et al. (2021), fennel EO showed an acaricidal efficacy of less than 30% [40]. However, in this case, two considerations must be made. The first one concerns the techniques used to verify the pharmacological action. Whereas in our study the action of vapours was investigated, in the aforementioned work acaricidal activity by contact was tested. If analyzed with attention, the results obtained with the EOs extracted from the same botanical species are not comparable. In fact, different techniques were used that affected the toxicity of the molecules constituting the phytocomplex. Differences in efficacy may also be related to variations in components. The EOs of the same plant may vary in composition depending on the soil in which the plants of origin are grown, temperature, harvest time, extraction methods, etc [41]. These conditions imply that the biological activities of the EOs extracted from the same plant species may differ due to variations in chemical composition, which may influence the synergies and antagonisms generated between the molecules. In biological contact assays, it is easy to understand that toxic compounds must penetrate directly into the cuticular layer of the arthropod to exert their action. In biological fumigation assays, toxic compounds are inhaled. Therefore, EOs with higher amounts of lipophilic and viscous compounds may perform better in biological contact tests. In biological fumigation assays, inhaled vapours act on various receptor targets in the nervous system. Vapour pressure also influences the effectiveness of a compound, as do the concentrations of the volatile parts. For example, in a study in which the vapours of different EOs were tested in open and closed chambers, it was shown that the most effective of the three oils tested (manuka EO) caused only 30% mortality in an open chamber. The same tests carried out in a closed chamber resulted in a mortality of >80% [42]. The same indications can be deduced from our previous experiment, in which we tested the effectiveness of oregano EO using a closed-pipe fumigation system. With this system, with just 1 mg/mL we obtained high mite mortality [11]. In this research work, using a semi-field test, we had to use an EO concentration 20 times higher to achieve an appreciable efficacy. Surely, the test was also affected by the size of the cage: the volume of air to be saturated was greater than in the Eppendorf. In conclusion, we must state that fennel EO proved to be a moderately effective tool for controlling the *V. destructor* parasite. However, its effectiveness is lower than that of the oregano EO tested in our previous study [11].

Therefore, the use of fennel EO can be considered in an integrated pest management program (IPM), in rotation with other molecules. EOs, in fact, have characteristics that make them particularly attractive in this context. EOs are low-risk products: from an ecotoxicological point of view, their use does not present any potential problems. For example, unlike other synthetic insecticides, the components of EOs are biodegradable and have short half-lives, and so far no biomagnification has been documented [43,44].

One limitation is the variable effectiveness in experiments conducted even with the same EO [45]. This variability is a problem of no small importance for possible commercial applications. In this regard, it is important to investigate the phytochemical profile of the oil in question to better define the anti-parasitic action. Despite these advantages, the marketing of essential-oil-based drugs should only take place after a careful analysis of toxicity to non-targets. The handling of these compounds is poor, and often the toxic dose for honeybees is close to that for mites. In the toxicity tests, the EOs showed toxic effects. The honeybees writhed in convulsions and spasms that led to death when they were exposed to high dosages. Therefore, if not used in the right doses, these EOs can cause mortality in colony individuals [23,46]. It is therefore recommended to use these EOs with caution to control *V. destructor* in bee colonies. Certainly, the high volatility makes them difficult to handle. Therefore, comprehensive laboratory studies are required to establish effective concentrations/doses and to devise delivery systems that are not affected by environmental conditions of temperature and humidity. The advantage of using a semi-field test lies in the possibility of calculating the volume of air present in the experimental cage and relating it to the volume present inside the hive. The air volume inside the cage in our experiments was one litre, whereas the air in a Dadant–Blatt hive is 60 litres. Given the relevant proportions, the volume needed to obtain the same pharmacological effect in vivo can then be calculated. In conclusion, this study allowed us to establish that semi-field studies are indispensable and must necessarily accompany laboratory tests on EOs that have greater control potential. Semi-field tests are an intermediate step between in vitro and in hive investigations. On the other hand, in vitro tests can be considered for preliminary screenings to determine whether an EO may have pest control potential.

## 5. Conclusions

Synthetic drugs show an increasing loss of effectiveness due to the emergence of resistance phenomena reported in several animal species [6,7,47,48,49,50]. An alternative to synthetic acaricides is the use of EOs, which are an extremely promising tool. These secondary metabolites of plant species, in fact, have a varied composition that makes it difficult for treated pest populations to develop drug resistance. Furthermore, with a view to One Health, it is important to underline that EOs also have a low toxicity for humans and a minimal environmental impact. Their use is therefore extremely favourable. This study paves the way for this mindset. In fact, semi-field and field efficacy studies are scarce. All in all, it must be said that this research work is part of that important branch of veterinary medicine that goes by the name of “Green Veterinary Pharmacology”, which will develop more and more over the years.

## Figures and Tables

**Figure 1 vetsci-09-00684-f001:**
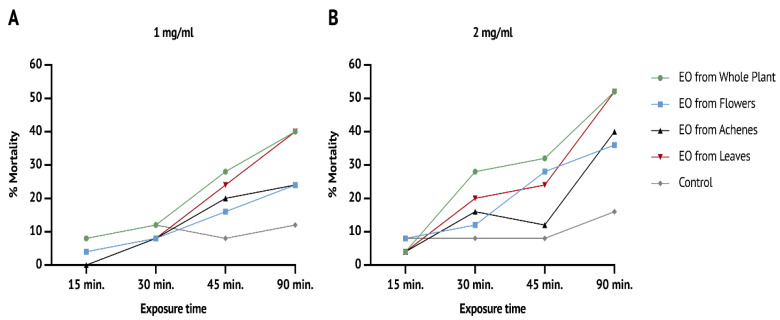
Mortality rates reached after 15, 30, 45 and 90 min of exposure to EO vapours at (**A**) 1 mg/mL concentration and (**B**) 2 mg/mL concentration.

**Figure 2 vetsci-09-00684-f002:**
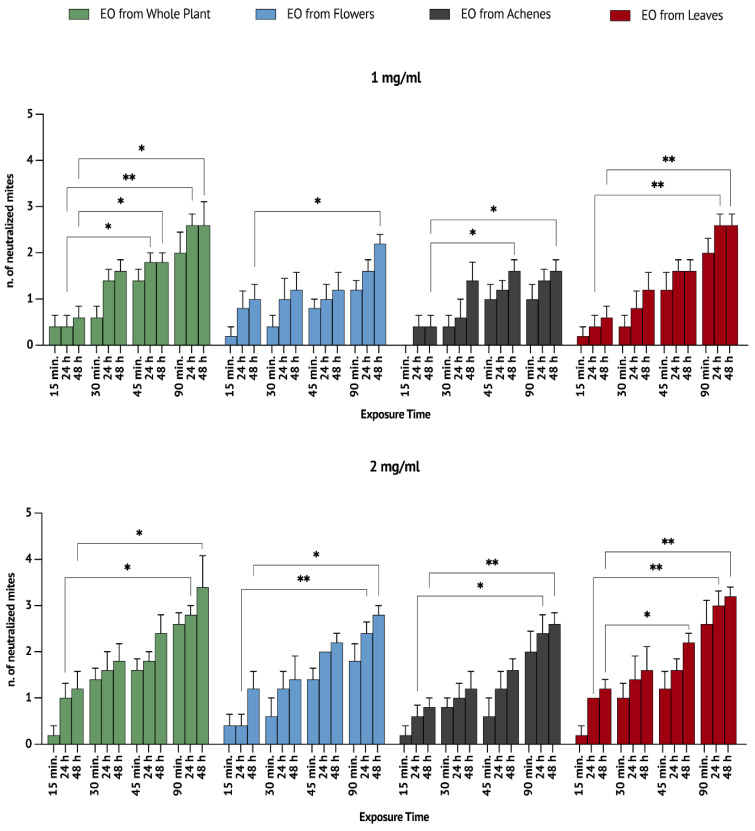
Number of neutralized mites after the incubation of the surviving mites with a honeybee larva (as nourishment) after 24 and 48 h of exposure to EO vapours at the concentrations of (**above**) 1 mg/mL and (**below**) 2 mg/mL. The results are expressed as means ± SEMs. * *p* < 0.05, ** *p* < 0.01.

**Figure 3 vetsci-09-00684-f003:**
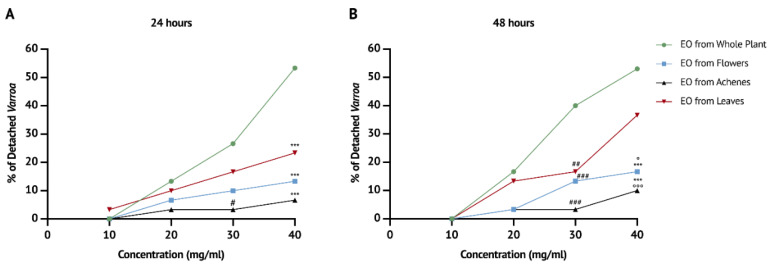
Mortality percentages reached after (**A**) 24 h and (**B**) 48 h of exposure to EO vapours. *** *p* < 0.001 vs. EO from whole plant 40 mg/mL; # *p* < 0.05, ## *p* < 0.01, ### *p* < 0.001 vs. EO from whole plant 30 mg/mL; ° *p* < 0.05; °°° *p* < 0.001 vs. EO from leaves 40 mg/mL.

**Table 1 vetsci-09-00684-t001:** Phytochemical profiles of *Foeniculum vulgare* subsp. *piperitum* essential oils.

Compound ^(a)^	RT ^(b)^	RAP ^(c)^			
		Whole plant	Leaves	Achenes	Flowers
α-pinene	6.585	-	2.50 ± 0.23	1.71 ± 0.03	3.45 ± 0.30
β-myrcene	7.774	-	-	0.41 ± 0.02	1.13 ± 0.02
o-cymene	8.070	0.92 ± 0.02	-	-	-
α-phellandrene	8.111	5.72 ± 0.09	8.37 ± 0.17	1.53 ± 0.03	6.10 ± 0.37
p-cymene	8.471	-	-	0.20 ± 0.02	0.91 ± 0.04
Limonene	8.569	-	5.54 ± 0.28	5.67 ± 0.34	6.05 ± 0.22
Eucalyptol	8.609	-	-	0.10 ± 0.02	0.70 ± 0.03
β-ocimene	8.672	-	0.20 ± 0.02	0.10 ± 0.01	1.10 ± 0.11
γ-terpinene	9.032	-	0.70 ± 0.03	1.1 ± 0.01	2.5 ± 0.12
Linalool	9.250	0.69 ± 0.02	-	-	-
Fenchone	9.649	7.05 ± 0.24	9.76 ± 0.88	18.48 ± 1.28	16.08 ± 1.02
Camphor	10.438	-	0.22 ± 0.03	1.64 ± 0.08	1.03 ± 0.08
p-anisaldehyde	11.110	5.49 ± 0.10	-	-	-
Estragole	11.129	5.00 ± 0.08	30.83 ± 2.22	33.40 ± 2.15	28.81 ± 2.04
Isoeugenol	12.070	0.73 ± 0.01	-	-	-
Anethole	12.232	49.90 ± 0.98	29.18 ± 1.42	24.16 ± 1.85	27.40 ± 1.93
α-bergamotene	12.630	0.74 ± 0.02	-	-	-

*^(^*^a)^ Main compounds listed in order of elution from MS column; ^(b)^ Retention time (as min); ^(c)^ Relative area percentage (peak area relative to total peak area in total ion current (TIC)%). Each value is the mean ± SD of three independent measurements.

**Table 2 vetsci-09-00684-t002:** Mortality percentages obtained after 15, 30, 45 and 90 min of exposure to EO vapours.

Time of Exposition	EO from Whole Plant		EO from Flowers		EO from Achenes		EO from Leaves	
	1 mg/mL	2 mg/mL	1 mg/mL	2 mg/mL	1 mg/mL	2 mg/mL	1 mg/mL	2 mg/mL
15 min	8%	4%	4%	8%	0%	4%	4%	4%
30 min	12%	28%	8%	12%	8%	16%	8%	20%
45 min	28%	32%	16%	28%	20%	12%	24%	24%
90 min	40%	52%	24%	36%	24%	40%	40%	52%

**Table 3 vetsci-09-00684-t003:** Mortality percentages related to subacute toxicity at (above) 1 mg/mL concentration and (below) 2 mg/mL concentration.

**Concentration** **1 mg/mL**	**EO from** **Whole Plant**		**EO from** **Flowers**		**EO from** **Achenes**		**EO from** **Leaves**	
	**24 h**	**48 h**	**24 h**	**48 h**	**24 h**	**48 h**	**24 h**	**48 h**
15 min	8%	12%	16%	20%	8%	8%	8%	12%
30 min	28%	32%	20%	24%	12%	28%	16%	24%
45 min	36%	36%	20%	24%	24%	32%	32%	32%
90 min	52%	52%	32%	44%	28%	32%	52%	48%
**Concentration** **2 mg/mL**	**EO from** **Whole Plant**		**EO from** **Flowers**		**EO from** **Achenes**		**EO from** **Leaves**	
	**24 h**	**48 h**	**24 h**	**48 h**	**24 h**	**48 h**	**24 h**	**48 h**
15 min	20%	24%	8%	24%	12%	16%	20%	24%
30 min	32%	36%	24%	28%	20%	24%	28%	32%
45 min	36%	48%	40%	44%	24%	32%	32%	44%
90 min	56%	68%	48%	56%	48%	52%	60%	64%

**Table 4 vetsci-09-00684-t004:** Detachment percentages registered after 24 and 48 h of exposure to EO vapours.

Concentration	EO from Whole Plant		EO from Flowers		EO from Achenes		EO from Leaves	
	24 h	48 h	24 h	48 h	24 h	48 h	24 h	48 h
10 mg/ml	0%	0%	0%	0%	0%	0%	3.33%	0%
20 mg/ml	13.33%	16.67%	6.67%	3.33%	3.33%	3.33%	10%	13.33%
30 mg/ml	26.67%	40%	10%	13.33%	3.33%	3.33%	16.67%	16.67%
40 mg/ml	53.33%	53.33%	13.33%	16.67%	6.67%	10%	23.33%	36.67%

## Data Availability

The data are kept at the University of Magna Græcia of Catanzaro and are available upon request.

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
