# Peer review of "Phytochemical Profile of Foeniculum vulgare Subsp. piperitum Essential Oils and Evaluation of Acaricidal Efficacy against Varroa destructor in Apis mellifera by In Vitro and Semi-Field Fumigation Tests"

_vetsci, 2022, doi:10.3390/vetsci9120684_

Round 1

Reviewer 1 Report

Varroa destructor has posed a great threat on honey bees. More efficient and safer methods are urgently needed to control this parasites. The current study analyzed the chemical profiles of fennel essential oils and evaluated their acaricidal efficacy against Varroa mites. The authors have gain some positive results that indicate the fennel EOs have potentials in controlling the mites. However, I have a few concerns on the manuscript including writing and language usage, experimental design, etc. My major and minor comments are listed below.

Major comments:

1.         There are many issues about the writing and language usage such as fragmented sentences, misnomer and so on, which often makes me confusing.

2.         The sampling of mites and bees are not described in detail:

L138. What kind of mites did you collected? All of the individuals you could see? The status of the test mites is quite important and affect the output of the following treatment, so it must be clearly stated.

L141-143. How did you identify the weak mites, and what standards did you use? In my opinion, it is impossible to efficiently screen the termed ‘weak’ mites. The more realistic way is to test the mites as soon as possible. For instance, use them with one hour.

L144-149. What were the source colonies for the honey bee sampling? Were they from the same colonies that used to raise Varroa mites? If so, the honey bees were not healthy even though they emerged without any mite on their body.

Minor comments:

3.         L40. Better use “caused” instead of “sustained”.

4.         L47. Delete “that isolated from” since it is a redundant description.

5.         L52. Should be “in vitro”.

6.         L55. Use simple past tense and keep it consistency.

7.         L57. Here, “instead” is incorrect. Consider “while”.

8.         L59. Should be “increasing”.

9.         L60-62. The last sentence is too long to make readers confusing.

10.     L71. Better use “phoretic” to keep the same as in the cited paper (Ref. 3)

11.     L77. Should be “coumaphos”.

12.     L80. Ref. 5 and ref. 6 are not closely related to the statement.

13.     L83-86. Wrongly composed sentence. Please re-write it.

14.     L90-92. What within specialized cells or glands? Very confusing description.

15.     L95. Whose efficacy?

16.     L101. The expression duplicated by saying “the use of”.

17.     L110-112. What is the relationship between the two “to”?

18.     L136. The exact temperature and humidity should be provided.

19.     L137. Operculum? What’s that? Too many non-professional words! It’s required to use specialist vocabulary so that researchers in the same field can understand with each other. We should not deliberately pursue new words and use them in a relatively mature field.

20.     L146. Flicker? What does it mean?

21.     L164. In this section, the detailed method for GC-MS analyses including compound identification was missing. Please add it.

22.     L184. Fragmented sentence. Please re-write it.

23.     L191. The environmental conditions of the test must be clarified.

24.     L192. Please double-check the published year of ref. 20.

25.     L195. What kind of adult workers used here?

26.     L210. The toxicity evaluation of honey bee workers?

27.     L211.How old were the adult bees?

28.     L214. The Ref. 55 is missing.

29.     L232. The title of Table 1 is missing and the table was poorly organized.

30.     L260. After an exposure time of 24 h? No, you did not do that. Please double-check the description in the Materials and Methods section.

31.     L264. How to understand the ‘neutralized mites’? Were they dead or immobile?

32.     L279-282. It’s too long to understand the meaning.

33.     L310. ‘Mortality percentage’ used in the figure caption, but ‘% of detached’ used in figure body. Do they have the same meaning?

34.     L330. Pests.

35.     L332. ‘Not absent’ or ‘absent’?

36.     L343. Fragmented sentence.

37.     L368-369. References must be provided for this statement.

38.     L398-412. No references are needed for the whole paragraph? It seems not.

Author Response

Comments and Suggestions for Authors

Varroa destructor has posed a great threat on honey bees. More efficient and safer methods are urgently needed to control this parasites. The current study analyzed the chemical profiles of fennel essential oils and evaluated their acaricidal efficacy against Varroa mites. The authors have gain some positive results that indicate the fennel EOs have potentials in controlling the mites. However, I have a few concerns on the manuscript including writing and language usage, experimental design, etc. My major and minor comments are listed below.

Response: we sincerely thank the reviewer, whose careful reading enabled us to significantly improve the quality of our manuscript. Responses to major comments and minor revisions have been provided in the paper.

Major comments:

  1. There are many issues about the writing and language usage such as fragmented sentences, misnomer and so on, which often makes me confusing.

Response: Now the English language was extensively revised by a native speaker. Sentences that could be confusing have been reworded. The introduction was implemented and more bibliographical notes were added

  1. The sampling of mites and bees are not described in detail:

L138. What kind of mites did you collected? All of the individuals you could see? The status of the test mites is quite important and affect the output of the following treatment, so it must be clearly stated.

Response: thank you for this comment. As a result, additional information on the collected mites has been added to the manuscript at line 177.

L141-143. How did you identify the weak mites, and what standards did you use? In my opinion, it is impossible to efficiently screen the termed ‘weak’ mites. The more realistic way is to test the mites as soon as possible. For instance, use them with one hour.

Response: we thank the reviewer for the comment and we have deleted the term “weak” as suggested. However, in literature the term “weak” is widely used when mites with abnormal behavior are collected and, consequently, discarded.  Below you can find some articles

 1) Garedew, Assegid, Erik Schmolz, and Ingolf Lamprecht. "The energy and nutritional demand of the parasitic life of the mite Varroa destructor." Apidologie 35.4 (2004): 419-430;

2) Damiani, Natalia, et al. "Acaricidal and insecticidal activity of essential oils on Varroa destructor (Acari: Varroidae) and Apis mellifera (Hymenoptera: Apidae)." Parasitology research 106.1 (2009): 145-152.

L144-149. What were the source colonies for the honey bee sampling? Were they from the same colonies that used to raise Varroa mites? If so, the honey bees were not healthy even though they emerged without any mite on their body.

Response: we sincerely thank the reviewer for this observation, which helped to improve the quality of the manuscript. The honeybees used in the experimental trials belonged to healthy hives from apiary previously treated for Varroa mite. The mites were collected from apiary not treated with acaricides. This information has been added to the text at line 273, 274.

Minor comments:

  1. L40. Better use “caused” instead of “sustained”.

Response: is now changed

  1. L47. Delete “that isolated from” since it is a redundant description.

Response: deleted

  1. L52. Should be “in vitro”.

Response: corrected as suggested

  1. L55. Use simple past tense and keep it consistency.

Response: corrected as suggested

  1. L57. Here, “instead” is incorrect. Consider “while”.

Response: corrected

  1. L59. Should be “increasing”.

Response: corrected

  1. L60-62. The last sentence is too long to make readers confusing.

Response: the sentence was divided into two periods

  1. L71. Better use “phoretic” to keep the same as in the cited paper (Ref. 3)

Response: we prefer to maintain “dispersal” and change the bibliographic note with another more recent

  1. L77. Should be “coumaphos”.

Response: corrected

  1. L80. Ref. 5 and ref. 6 are not closely related to the statement.

Response: we thank the reviewer for this observation. In the text of the aforementioned articles, the authors frequently mention drug resistance and deal with it. The phenomena of drug resistance are the pretext for the study about the essential oil treatments in the first article. In the second article, drug resistance is referred to in the entire abstract and the conclusion of the review deals about drug resistance phenomena. We believe that both are congruent to explain the concept and we would prefer to keep them. If you consider them unsuitable at all, we will replace these mentioned items with others.

  1. L83-86. Wrongly composed sentence. Please re-write it.

Response: the sentence has been rephrased

  1. L90-92. What within specialized cells or glands? Very confusing description.

Response: we have reconstructed the sentence explaining the concept better

  1. L95. Whose efficacy?

Response: the sentence has been rewritten and a bibliographic note has been added.

  1. L101. The expression duplicated by saying “the use of”.

Response: corrected as suggested

  1. L110-112. What is the relationship between the two “to”?

Response: the sentence has been rephrased to clarify the concept

  1. L136. The exact temperature and humidity should be provided.

Response: the information was added

  1. L137. Operculum? What’s that? Too many non-professional words! It’s required to use specialist vocabulary so that researchers in the same field can understand with each other. We should not deliberately pursue new words and use them in a relatively mature field.

Response: thank you for this advice. However, the term “operculum” is commonly used in scientific literature of the field. Below, you can find some references:

1) Diaz, Tsiri, et al. "Alterations in honey bee gut microorganisms caused by Nosema spp. and pest control methods." Pest management science 75.3 (2019): 835-843;

2) Soares, Michelle PM, et al. "A cuticle protein gene in the honeybee: expression during development and in relation to the ecdysteroid titer." Insect biochemistry and molecular biology 37.12 (2007): 1272-1282

For this reason, with your agreement, we would prefer to leave this term in the text.

  1. L146. Flicker? What does it mean?

Response: the sentence was rephrased and the term has been replaced

  1. L164. In this section, the detailed method for GC-MS analyses including compound identification was missing. Please add it.

Response: Many thanks, the details of the method for GC-MS analyses has been added as requested.

  1. L184. Fragmented sentence. Please re-write it.

Response: the sentence was rephrased as suggested

  1. L191. The environmental conditions of the test must be clarified.

Response: the required information has been added

  1. L192. Please double-check the published year of ref. 20.

Response: we have corrected

  1. L195. What kind of adult workers used here?

Response: details about the adults were added to the text

  1. L210. The toxicity evaluation of honey bee workers?

Response: in the paragraph, additional information was added

  1. L211.How old were the adult bees?

Response: the information is detailed at the beginning of the materials and methods (“the honeybees used for the tests came from a now nascent brood honeycomb that was transported to the lab, where it was placed in an incubator. In the incubation conditions, the adults had the possibility to come out”)

  1. L214. The Ref. 55 is missing.

Response: we have corrected

  1. L232. The title of Table 1 is missing and the table was poorly organized.

        Response: the title has been added and the whole table has been reorganized as requested.

  1. L260. After an exposure time of 24 h? No, you did not do that. Please double-check the description in the Materials and Methods section.

Response: we thank the reviewer for this very important comment. Mites were not exposed to the vapors for 24 hours nor for 48 hours. Our objective was to check for any chronic toxicity due to the initial exposure. We corrected in accordance with the suggestion.

  1. L264. How to understand the ‘neutralized mites’? Were they dead or immobile?

Response: many thanks for this observation. The mites were classified as neutralized when they didn’t walk but move one o more leg. The information was added to the text

  1. L279-282. It’s too long to understand the meaning.

Response: the sentence was divided into two periods

  1. L310. ‘Mortality percentage’ used in the figure caption, but ‘% of detached’ used in figure body. Do they have the same meaning?

Response: we modified and used the same nomenclature in both cases

  1. L330. Pests.

Response: corrected

  1. L332. ‘Not absent’ or ‘absent’?

Response: the sentence has been rephrased to clarify the concept

  1. L343. Fragmented sentence.

Response: the sentence was corrected in accordance to the suggestion

  1. L368-369. References must be provided for this statement.

Response: added

  1. L398-412. No references are needed for the whole paragraph? It seems not.

Response: references were added as suggested

Reviewer 2 Report

Dear Authors,

It's a good MS, I like it. It is of great interest to the scientific community involved in varroa control. It specifies methods that can be replicated and standardized in some way to study other essential oils. I have a series of comments/suggestions for you to assess and clarify.

It would be very positive if in material and methods the authors added some scheme, drawing, or photography. For example the starting plant material (its parts) ... the housing of bees and insects ... the steps of the studies carried out in M&M with their specifications (in the form of a figure/diagram/scheme).

Concerning the chemical profile, the data on the concentration of the compounds must be presented in the results. Don't have the validation parameters for the analytical method? They must be attached (calibration line, detection limits). The units must be specified in the table.

L164 2.2 Extraction techniques and phytochemical profile EOs were obtained from fresh samples, which were washed and extracted with steam distillation for 2 h.

- What amount of plant (by weight) of each part is taken for the distillation process? What is the yield obtained from essential oil starting from that vegetable mass taken? .

Table 2 and Table 3, remove ".00".

The figures are too small and difficult to read. Could the authors increase the size/resolution?

The conclusions must be concise, with some significant numerical data obtained in this work. The authors should completely rewrite this section. This section should not be referenced. Some part of the current conclusions is more suitable in results and discussions.

Author Response

Reviewer 2

Dear Authors,

It's a good MS, I like it. It is of great interest to the scientific community involved in varroa control. It specifies methods that can be replicated and standardized in some way to study other essential oils. I have a series of comments/suggestions for you to assess and clarify.

It would be very positive if in material and methods the authors added some scheme, drawing, or photography. For example, the starting plant material (its parts) ... the housing of bees and insects ... the steps of the studies carried out in M&M with their specifications (in the form of a figure/diagram/scheme).

Response: we sincerely thank the reviewer for his appreciation of the manuscript. Unfortunately, we did not take any photos during the experimental phases. We will follow this advice and take pictures in the next research work.

Concerning the chemical profile, the data on the concentration of the compounds must be presented in the results. Don't have the validation parameters for the analytical method? They must be attached (calibration line, detection limits). The units must be specified in the table.

Response: results have been discussed in the Results section. The details of the method for GC-MS analyses have been added to the Materials and Methods section (paragraph 2.2). We did not perform a quantitative analysis, but a qualitative analysis in which the relative abundance of each compound has been expressed in RAP (Relative area percentage), the peak area relative to total peak area in total ion current (TIC) % (as specified in the table footer). Results were reported as the mean ± S.D. of three independent measurements.

L164 2.2 Extraction techniques and phytochemical profile EOs were obtained from fresh samples, which were washed and extracted with steam distillation for 2 h.

- What amount of plant (by weight) of each part is taken for the distillation process? What is the yield obtained from essential oil starting from that vegetable mass taken? .

       Response: the amount of plant material and the yields obtained have been reported in the revised version of the manuscript.

Table 2 and Table 3, remove ".00".

Response: done

The figures are too small and difficult to read. Could the authors increase the size/resolution?

Response: we thank the reviewer for the advice. Image resolution is maximum. To meet your kind request, we have increased the size of the image.

The conclusions must be concise, with some significant numerical data obtained in this work. The authors should completely rewrite this section. This section should not be referenced. Some part of the current conclusions is more suitable in results and discussions.

Response: many thanks to the reviewer for this important advice. Accordingly, the section was rewrite

Round 2

Reviewer 1 Report

No more comments and suggestions for authors.